# Improving Sonication Efficiency in Transcranial MR-Guided Focused Ultrasound Treatment: A Patient-Data Simulation Study

**DOI:** 10.3390/bioengineering11010027

**Published:** 2023-12-26

**Authors:** Changsoo Kim, Matthew Eames, Dong-Guk Paeng

**Affiliations:** 1Research Institute for Basic Sciences, Jeju National University, Jeju 63243, Republic of Korea; yustchangjnu@jejunu.ac.kr; 2Focused Ultrasound Foundation, Charlottesville, VA 22903, USA; matt.d.c.eames@gmail.com; 3Department of Radiology, University of Virginia, Charlottesville, VA 22903, USA; 4Ocean System Engineering, Jeju National University, Jeju 63243, Republic of Korea

**Keywords:** MR-guided focused ultrasound, sonication efficiency improvement, transducer tilting

## Abstract

The potential improvement in sonication efficiency achieved by tilting the focused ultrasound (FUS) transducer of the transcranial MR-guided FUS system is presented. A total of 56 cases of patient treatment data were used. The relative position of the clinical FUS transducer to the patient’s head was reconstructed, and region-specific skull density and porosity were calculated based on the patient’s CT volume image. The total transmission coefficient of acoustic waves emitted from each channel was calculated. Then, the total energy penetrating the human skull—which represents the sonication efficiency—was estimated. As a result, improved sonication efficiency was by titling the FUS transducer to a more appropriate angle achieved in all 56 treatment cases. This simulation result suggests the potential improvement in transcranial-focused ultrasound treatment by simply adjusting the transducer angle.

## 1. Introduction

Transcranial-focused ultrasound (TcFUS) using a largely phased array transducer has become an attractive modality to treat many brain diseases. Initial clinical trials have reported the treatment of brain tumors [1], neuropathic pain [2,3], essential tremor [4,5], Parkinson’s disease [6,7], and blood–brain barrier opening in patients with Alzheimer’s disease [8] and brain tumors [9]. After its initial research [10,11], the ExAblate Neuro TcFUS system developed by Insightec (Haifa, Israel) is both commercially available and approved by the U.S. Food and Drug Administration (FDA) for the clinical treatment of essential tremor and Parkinson’s disease [12]. The ExAblate Neuro consists of a hemispherical transducer for emitting high-power therapeutic ultrasound using 1024 individually controllable transducer elements (phased array). Such phased arrays can steer the focal point electronically by modulating the phase and amplitude of the individual driving signals to compensate the skull-induced aberration [13].

The human skull is a complex medium composed of heterogeneous and viscoelastic structures, which is a major challenge to FUS brain applications. In the process of ultrasound transmission through the skull to reach a targeted spot, unwanted transmission loss with refraction and absorption of ultrasound energy occurs. A parameter called skull density ratio (SDR) or skull score (SS), i.e., the ratio of the Hounsfield unit value of the marrow layer to the one of the cortical layer on cranial computed tomography (CT), is used to estimate the absorption of ultrasound energy on the skull layer. A low SDR value (<0.4) is correlated with greater loss of ultrasound energy by the skull [14]. Another important cause of ultrasound energy degradation is transmission loss occurring on the boundaries between two different materials, such as the water-skull layer and skull-brain tissue layer. The transmission loss varies greatly depending on the incident angle of the ultrasound beam relative to the skull surface [15,16]. The largest possible incident angle to allow the wave to penetrate the boundary should always be smaller than the critical angle, and any wave with an incident angle greater than the critical angle is completely reflected. For this reason, the channels in the phased array with an incident angle greater than the critical angle were usually turned off in the clinical system to avoid unnecessary heat generation to the skull and to avoid the generation of unpredictable shear waves. Due to non-spherical skull calvaria and variation of skull composition where the beam encounters the skull, the total count of active ultrasound elements also varies depending on the patient and selected intracranial target (focal point) as well as the placement angle of transducer related to the patient head. In other words, an optimum transducer placement could be achieved by placing the transducer at an angle that produces the most efficient transmission of therapeutic ultrasound through the patient’s skull, which eventually leads to improved sonication. Several clinical studies have shown that some of the treatment cases failed to achieve lesioning via TcFUS because of insufficient temperature rise at the focal point [3,17,18]. Since the ExAblate system is already equipped with a practical phase and amplitude compensation module to maximize the focal pressure, the conventional approach for focal pressure improvement based on skull aberration correction may not be effective in these aborted treatment cases. However, the improvement in sonication efficiency by placing the transducer in the optimum position was not attempted.

The essential means to estimate the improvement in the sonication efficiency of a newly proposed approach is through computational simulation. The patient-scanned CT image-based simulation could be an effective tool for the estimation of sonication efficiency. Several different simulation methods, including finite differences time domain (FDTD) method [19,20], k-Space [21,22,23], Hybrid Angular Spectrum (HAS) [24,25,26], and raytracing [27,28,29,30], were used to calculate skull-induced aberration of ultrasound wave to estimate focal pressure and refocusing of ultrasound energy. However, none of these simulation studies attempted to simulate the optimum transducer positioning for improved sonication efficiency. 

This study introduces a sonication efficiency estimation method by considering the skull porosity, region-specific skull density, and incident angle. A series of simulations were executed to determine the optimum transducer placement that had the highest sonication efficiency. A total of 56 patient treatment data were used in this study to validate if the sonication efficiency could be improved by tilting the transducer. As a result, the transducer tilting could improve the sonication efficiency in all the cases studied, which suggests a potentially applicable method in the current clinical setup for patient treatment.

## 2. Materials and Methods

### 2.1. Raytracing Implementation

The raytracing algorithm implemented in Kranion [30] was used in this study. Kranion is a treatment data visualization software developed by the Focused Ultrasound Foundation (Charlottesville, VA, USA). Briefly, an initial ray that represents the direction of a plane wave from each transducer element was constructed by connecting the transducer coordinate and sonication targeted point (Figure 1a). Then, the collision point between the ray and skull outer table was detected, and Snell’s law was implemented to calculate the transmission coefficient and the first refracted ray. Similarly, another collision point between the first refracted ray and the inner skull table was detected. The second refracted ray was calculated by using the same method. Finally, the overall transmission loss of each channel was estimated and validated with tissue heating efficiency which is explained in detail in the later sections. The block diagram of the overall raytracing process is shown in Figure 1b and an example of the raytracing result through the water-skull-brain layer is illustrated in Figure 1c.

#### 2.1.1. Collecting Voxel Value along the Ray

An algorithm called the 3D digital differential analyzer (3D-DDA) was used to calculate the integer coordinate along a line in 3D, which is defined by two 3D points [31]. To calculate the integer coordinate, the start and end points of the 3D line should be defined before calculation. The start point was defined as the transducer coordinate, and the endpoint was set as the sonication target point. Then, the 3D vector could be defined by these two points. As shown in Figure 1d, the integer coordinate of the voxels along the defined line could be calculated. Since the coordinate of each voxel along the 3D line was defined, it allows the assessment of the voxel value along the ray, which is the Hounsfield Unit (HU).

#### 2.1.2. Neighbor Averaging-Based Voxel Value Interpolation

Since the medical volume image is normally stored in a finite cubic space, it forms a gap between two layers or even two voxels. When collecting the voxels along a defined trajectory, a coarse profile could be unavoidable. A neighbor voxel averaging method was used to gain a smoother voxel profile along the ray. A total of 26 neighbor voxel values were collected at each voxel coordinate of the ray. Then, an average of 26 neighbor voxel values was calculated and defined as new voxel values. By repeating this process on all voxels along the ray, a smoother voxel profile could be obtained, as shown in Figure 1d. Note that it is recommended to use the neighbor averaging method rather than other signal processing, such as bandpass filtering or interpolation, because the resulting profile from another method will largely depend on the parameters used during the smoothing process. Since the shape and composition of the skull layer vary in different regions, other filtering methods cannot robustly remove the noise from the profile. In some extreme cases, another filtering method may introduce unexpected alterations to the signal, and this will confuse the estimation process. On the contrary, the neighbor averaging method does not require any parameter and relies on the data itself. Thus, it can robustly smooth the HU profile along the ray.

#### 2.1.3. Skull Solidity Calculation

As shown in Figure 1e, the solidity of the local skull along the ray was assumed to be the ratio between the voxel intensity area and the filled intensity area. To derive the solidity, firstly, two peaks (black dots in Figure 1e) indicate the start and end of skull boundaries, which were located on the skull density profile (red curve in Figure 1e) along each ray path. Then, the area of skull data (dashed line marked area) and the area of filled skull data (yellow rectangle) were calculated. Finally, the ratio between the two areas was derived and defined as the solidity of the skull.

#### 2.1.4. HU Data-Based Skull Density and Velocity

The acoustic properties of the skull, such as the sound velocity and the skull density, are essential to simulate the acoustic transmission through the skull layer. In this study, we derived the skull density from the CT data and calculated the sound speed of the skull based on the derived skull density [15]. The following equation depicts the conversion between HU value and density.
(1)$$\rho =\rho_{min}+\left(\rho_{max}-\rho_{min}\right)\frac{\bar{HU}-\bar{{HU}_{min}}}{\bar{{HU}_{min}}-\bar{{HU}_{max}}}$$
where the $\rho_{min}$ and $\rho_{max}$ are the maximum and minimum skull density in kg/m^3^ and $\bar{{HU}_{min}}$ and $\bar{{HU}_{max}}$ are minimum and maximum HU of the skull on CT. 

Then, the sound velocity of the skull was derived by implementing a genetic algorithm optimization, which was validated via experimental measurement [32]. The non-linear relationship between density and velocity resulted, as shown in Figure 1f.

#### 2.1.5. Collision Point Detection

The collision point detection is essential for implementing the refraction angle calculation in raytracing. A threshold-based collision point detection was used. The histogram of the voxel value from the CT volume was studied to define an optimum threshold. As shown in Figure 2, the histogram of the voxel intensity depicts the distribution of the CT intensity. The original CT image (Figure 2b) could turn into a binary image (Figure 2c–e) after applying a thresholding process. Briefly, the voxel with a value satisfying the defined threshold window was defined as foreground (white), and the other voxel not satisfying the threshold was set as background (black). The optimum threshold represents the boundary between the skull and tissue, which could be searched by repeating the thresholding process using different threshold combinations. A fine adjustment of the best threshold based on the histogram was discovered and used to remove the brain and scalp region. As a result, a threshold combination of 1155 and 3000 could remove the tissue region effectively, and this value was used as the threshold in the later collision detection process.

The voxel coordinate that satisfies the skull layer threshold from each ray was collected and defined as the collision point. As shown in Figure 1c and Appendix A, a distribution of collision points can be seen from this result, and a good matching between the collision point and the skull layer in the CT slice image confirms the effectiveness of collision point detection.

#### 2.1.6. Normal Vector Calculation

The collision points that depict the skull boundary could be utilized as an anchor point to search the normal vector of the skull boundary. Thus, the normal vector of the skull surface could be calculated by using the 5 × 5 × 5 edge operator introduced by Zucker and Hummer [33]. As shown in Appendix A, the normal vector (green line) could be derived by applying the edge operator to the collision point. The edge operator will calculate the gradient component along each axis, and the final normal vector could be derived by combining the gradient of each axis. Since the edge operator calculates the biggest gradient along each axis, it always returns a vector as a result. The result of normal vector calculation is highly dependent on where its basis point (the collision points) is located.

#### 2.1.7. Refraction Angle Calculation and Full Raytracing

The refraction angle on two transmission layers was calculated by using Snell’s law. Firstly, the incident angle of the wave was calculated after the surface normal vector was derived from the collision point. The angle between the ray segment, which connects the transducer coordinate to the first collision point, and the normal vector were calculated and defined as the incident angle. Then, Snell’s law was used to calculate the refraction angle by using the predefined speed of sound and density in water. However, channel-specific speed of sound and density on the skull layer were derived from the CT image. The second collision point, which is located on the surface of the inner skull table, could be detected by implementing the same collision point detection. Then, another normal vector should be derived based on the second collision point. Note that the direction of the second normal should be flipped after applying the edge operator because the voxel value becomes decent on the skull-to-brain layer. Similarly, the incident angle and refraction angle could be calculated by using the same methods.

#### 2.1.8. The Transmission Loss of Focused Ultrasound Energy

The major parameters contribute to acoustic energy loss, while transcranial propagation causes transmission loss through the water-to-skull layer and skull-to-brain layer and absorption of sound by the in-skull layers. The overall workflow of transcranial raytracing implemented in this study is shown in Appendix A. To make the realization of the total transmission loss, the raytracing calculation through skull layers (outer and inner skull tables) was implemented first. Then, the incident and refracted angles of each acoustic beam could be calculated. The transmission coefficient on a cortical bone layer could be derived by implementing Snell’s law as shown in the following equation.
(2)$$T_{p}(\theta )=\frac{2Z_{a2}cos\theta_{i}}{Z_{a2}cos\theta_{i}+Z_{a2}cos\theta_{t}},\;Z_{a1}=\rho_{1}c_{1},Z_{a2}=\rho_{2}c_{2}$$
where $\theta_{i}$, $\theta_{t}$, ρ and c are the incident angle, refraction angle, density, and sound speed, respectively. The transmission coefficient of the channel, which has an incident angle greater than the critical angle, was ignored in this study. 

The empirical relationship between skull solidity and absorption [19] was used to derive the absorption coefficient, and the following equation depicts the empirical relationship.
(3)$$A_{p}=\left(a_{min}+\left(a_{max}-a_{min}\right)\times \mu^{\beta }\right)\times t$$
where $a_{max}$, $a_{min}$, μ, β, and t indicate the empirical maximum absorption, empirical minimum absorption, skull solidity, absorption constant, and skull thickness, respectively. The maximum and minimum absorptions of the skull are defined as 0.2 and 8 dB/mm, respectively. A value of 0.5 was used as the absorption constant in this study, as suggested by [19]. Implementing this process on all the channels allows us to calculate each channel-specific absorption coefficient. 

Finally, the total transmission loss of focused ultrasound by skull was defined as the following equation:(4)$$T_{total}=T_{ws}\times (1-A_{p})\times T_{sb}$$
where $T_{total}$, $T_{ws}$, and $T_{sb}$ indicate the total transmission coefficient, transmission coefficient on the water-skull layer, and transmission coefficient on the skull-brain layer.

#### 2.1.9. Treatment Difficulty Score

A new parameter called treatment difficulty score (TDS) is defined to score the difficulty of FUS treatment. This parameter should consider the following factors. The first one is the heating efficiency, which is calculated by dividing the temperature change by sonication energy. The temperature change and sonication energy could be derived directly from the patient’s treatment record. The treatment has higher heating efficiency when less FUS sonication power is required to heat the brain tissue to meet the treatment goal. The second factor is the deviation of heating efficiency value over one treatment session. The smaller deviation of heating efficiency indicates that stable heating was performed. The third one is the total count of aborted sonications. Aborting a sonication is necessary for safety matters during sonication. However, it should be minimized to avoid elongated treatment time and reduce unwanted heating caused by unsuccessful sonications. A treatment with fewer aborted sonications indicates a smoother treatment flow and a more successful treatment. The last factor is the count of total sonications. The increased sonication count is required for stepwise heating in the treatment case with inefficient focal temperature rise. A weighted function was used to combine all four factors to represent TDS as Equation (5).
(5)$$TDS=W_{1}\times norm\left({heating\;efficiency}_{mean}\right)+W_{2}\times norm\left({heating\;efficiency}_{std}\right)+W_{3}\times norm\left(aborted\;sonication\right)+W_{4}\times norm\left(total\;sonication\right)$$
where the values of *W*_1_, *W*_2_, *W*_3_, and *W*_4_ were defined as 1, 0.25, 0.25, and 0.5 based on their contribution to the treatment difficulty, respectively. The TDS could be affected by the weight value. However, we did not attempt to investigate the impact of slight changes in these weight values on the TDS in this study.

#### 2.1.10. Total Delivered Sonication Energy

The single value to represent sonication efficiency was calculated by multiplying 100% on the total transmission coefficient, which includes the transmission coefficient on the inner and outer table of the skull and the absorption coefficient on the sponge skull layer. Since the sonication efficiency represents the averaged efficiency of each channel of a phased array, an additional parameter is required to account for the total amount of energy delivered to the target. To calculate the total amount of delivered sonication energy, the sonication efficiency was multiplied by the total used sonication channel. The sonication was terminated on the channel, which has an incident angle to the patient’s skull greater than the critical angle based on Snell’s Law to avoid the shear wave. 

Depending on the transducer positioning and its relative angle to the skull, the incident angle may change significantly, and the total number of active channels also changes. In the clinical setting, the amplitude and phase compensation were applied to each sonication channel to gain a clear focal spot. However, in this study, we assume a unified sonication power and sonication phase in all sonication channels. We assume the sonication could be improved by tilting the transducer to an optimum angle related to the patient’s head. To find the optimum transducer angle, the transducer was rotated on the *X*-axis (Frontal axis) and *Y*-axis with a predefined list of rotation angles, and the sonication efficiency and used sonication channel of each rotation were calculated. 

### 2.2. Patient Treatment Record Data

#### 2.2.1. Treatment Record Data

A total of 94 TcFUS clinical treatment exports were received from Virginia University Hospital [17,34,35] performed from the year 2015 to 2019 for Essential Tremor. However, only 56 of these treatment exports were compatible with the Kranion software used in this study. This study was approved by the University of Virginia Institutional Review Board (IRB-HSR# 21664), and all patients provided their written informed consent for participation.

#### 2.2.2. Patient Treatment Data Loading

To import the treatment data, the Kranion software is utilized to translate the raw treatment recording data to a MATLAB readable format (see Figure 3a). The information regarding the functionality of Kranioin software can be found on the Focused Ultrasound Foundation webpage [36]. The Kranion export contains a folder of image data and an XML file, including the 3D coordinates of each channel of the phased array, transducer-patient data registration, and all the attributes used in Kranion during visualization.

In Kranion, the CT and MRI image volumes, which were originally read from the patient treatment data, were translated to byte buffer format because it is the universal data format used on OpenGL-based rendering. To read the image volume correctly into MATLAB, the size of the image volume and the buffer format used to store the voxel value are required. This information can be found in the Kranion exported XML file.

The transducer coordinate information could also be found in the XML file, and the pattern of the transducer elements visualized in MATLAB is illustrated in Figure 3c. Note that the coordinate system of the transducer is different from the image coordinate system. An additional coordinate translation is required to combine the transducer information and image volume into the same coordinate system. The information on the transformation matrix can also be found in the XML file. Then, the raytracing (Figure 3d) based on patient treatment data could be implemented and the total transmission loss (Figure 3e) of focused ultrasound could be estimated.

### 2.3. Transducer Tilting Simulation

To estimate the optimum transducer angle, a series of simulations of the transducer positioned at each possible tilting angle should be performed. An overall sketch of transducer rotation and optimum rotation angle finding is illustrated in Figure 4. The notation to represent the rotation angle is defined in the form of (rotation angle on the frontal axis and rotation angle on the sagittal axis) as shown in Figure 4a–d. A tilting range of 15 degrees on both positive and negative directions for two rotation axes was used as the maximum limit for transducer tilting.

A 2D matrix of possible rotation angles for FUS transducer tilting was defined, and a combination of possible rotation angles on each rotation axis was rendered as a rectangular brick over the patient’s skull, as shown in Figure 4e. After the rotation matrix was defined, a series of simulations were executed, and the results of each case were reserved (Figure 4f). An illustration of a color-painted delivered sonication energy map is shown in Figure 4f. The rotation angle with the highest sonication efficiency, which is considered the optimum transducer positioning, is marked with a check sign in Figure 4f. Note that the illustration shown in Figure 4 is to explain the concept of the method. The actual transducer rotation implemented can be seen in Appendix A.

## 3. Results

### 3.1. Transducer Channel-Specific Skull Property

The channel-specific skull properties, such as HU, skull thickness, solidity, transmission coefficient, and absorption coefficient, are estimated as shown in Figure 5. The distribution of each parameter was rendered with a color map over the corresponding transducer coordinates.

### 3.2. Treatment Difficulty Score on 56 Patient Treatment Data

As shown in Figure 6 the TDS of all 56 treatments is the correlation value between heating efficiency, aborted sonication count, and total sonication count, and the TDS is further investigated, as shown in Table 1. Although the correlation value is low in each case, the sign of the correlation follows the expected trend. More specifically, increased heating efficiency could be expected in the case of decreasing aborted sonication and total sonication count cases. Conversely, a decreased heating efficiency could be expected in the case of increased aborted sonication count and increased total sonication count. In addition, an increase in aborted sonication count could be expected in the case of an increased total sonication count case.

### 3.3. Improvement in Delivered Sonication Energy by Transducer Tilting

The variation of sonication energy, activated sonication channel, and the delivered sonication energy (transmitted energy) by transducer tilting of two typical treatment cases is shown in Figure 7. A completely different distribution pattern of sonication energy and the used sonication could be seen. As the result shows, the tilted angle with the highest sonication efficiency does not necessarily mean a more activated sonication channel could be used on this sonication angle. The total delivered sonication energy accounts for both sonication efficiency and the used sonication channel, which could reveal the effectiveness of titling angle setting to the treatment.

A total of 56 patient treatment cases were simulated with the proposed transducer tilting. As Figure 8 shows, the transmitted energy could be improved in all the cases by titling the transducer to a higher position, and the average improvement in those patient treatment cases is 29.3 ± 13.8. Note that a decrease in sonication efficiency could also result in a particular rotation angle (red cross marker in Figure 7c,e) with an average decrease of sonication energy of about 39.6 ± 13.2.

## 4. Discussion

This study suggests the potential improvement in TcFUS sonication by titling the transducer to an optimum position. The raytracing-based sonication efficiency estimation implemented in this study could count the skull-specific density, porosity, and incident angle, which all significantly affect the transmission loss of therapeutic ultrasound through the skull bone. As the result suggests, an improved sonication could be achieved by tilting the transducer to a patient-specific optimum angle.

An experimental validation of the simulation result is desirable to validate the results further. A skull mimicking phantom or actual human cadaver skull could be utilized to diffuse the focal pressure. The CT scan of the skull phantom should be acquired before the validation experiment, and an initial sonication on the skull phantom with low power should be applied to generate the treatment recording data. The treatment recording data will serve as a key link between the physical setup and the simulation environment. An additional fixation jig allowing precise rotation of the transducer on both the Frontal and Sagittal axes is also required to physically rotate the skull phantom inside the clinical transducer. The experimental validation will be conducted in the next stage of the study in the near future.

This study is also bound by the following limitations. The scope of our investigation was limited to the thermal ablation treatment, dictated by the need for focal temperature measurement to evaluate the improved sonication. The potential to extend our method to nonthermal TcFUS treatment warrants further discussion. However, the sonication estimation model in predicting the sonication efficiency of the FUS transducer on a specific position related to the patient’s head could be applicable across a wider range of TcFUS treatments since the mechanism of ultrasound transmission through the skull remains the same. Furthermore, the absence of focal temperature estimation in the current model introduced potential challenges. In this simulation study, we excluded the focal temperature estimation, which could be derived by using the Pennes bioheat equation [37,38]. Since the focal pressure of TcFUS is required to implement this bioheat equation, it requires more computation sources and may not be feasible for the scene of the optimum transducer angle estimation, which requires dozens of simulation iterations.

Lastly, the sonication efficiency in this study was defined as the total transcranial ultrasound energy loss, including transmission losses that occurred on both the water-to-skull layer and skull-to-brain layer and the absorption loss caused by the sponge skull medium. Thus, it is different from the conventional full wave-based transcranial ultrasound simulation focused on focal pressure calculation, which usually requires a stepwise simulation on a uniformly divided sub-grid for pressure wave distribution, propagation, and transmission. Although it is believed that full-wave simulation outperformed compared to raytracing, the improvement in focal pressure by full-wave simulation compared to clinical raytracing software was observed within a 5% difference [15]. This could be one of the reasons that explain how the clinical FUS system could successfully treat lots of brain diseases in the past decade. However, the full-wave simulation becomes more accurate when the incident angle is greater than 20 digress because shear-wave propagation plays a significant role [39]. In the current study, we did not consider the shear wave because it does not significantly affect the energy accumulated in the focal area but the pattern of pressure distribution around the focal area [39]. Furthermore, the computation time is a crucial parameter in the translation of the proposed method to clinical setups. It becomes more significant to require a repetitive computation on a similar setup with a slight change of parameters. Since the study aims to illuminate the potential of transducer tilting to improve sonication efficiency in a clinical setup, we choose the raytracing model.

Despite these limitations, TDS accounts for multiple parameters to score the difficulty of TcFUS treatment. There is potential enhancement of this parameter through the incorporation of patient outcome-related parameters such as patient feedback during treatment and the post-treatment monitoring of tremor score. Moreover, an additional correlation study is required to explore the impact of each parameter and the paired weight function on TDS.

## 5. Conclusions

To summarize, our research underscores the potential of tilting the current clinical TcFUS transducer to an optimum position, which could improve the delivered sonication power. A fast and lightweight sonication estimation tool developed in this study could assist clinicians by providing an optimized transducer angle on the defined treatment target, consequently leading to more efficient sonication. Despite the limitations, our findings give insights into the potential improvements in TcFUS treatment, especially for patients who experienced inefficient heating on targeted tissue. The Matlab interface created for this project could also be used for several Insightec Neuro-specific simulations, including, for example, transducer optimal angle selection. It may also be easily applied to solve similar problems in a transducer-diagnostic scenario where alternative device geometries are imported or for simulations related to transducer device design optimization for TcFUS applications.

## Figures and Tables

**Figure 1 bioengineering-11-00027-f001:**
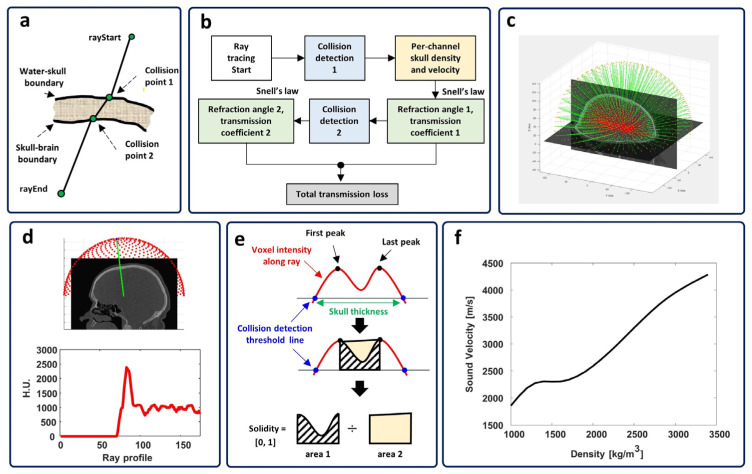
The implementation of raytracing. (**a**) the sketch of implemented raytracing on water-skull and skull-brain boundaries. (**b**) the block diagram of the overall raytracing process. (**c**) a typical example of a raytracing result. (**d**) the voxel value collection along the ray. (**e**) the solidity calculation method. (**f**) the relationship between skull density and sound velocity of skull derived from Hounsfield Unit.

**Figure 2 bioengineering-11-00027-f002:**
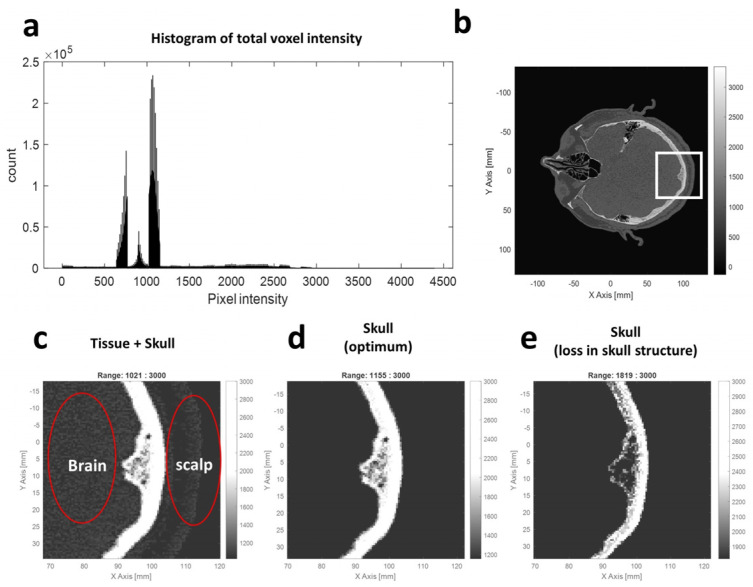
The threshold for collision point on skull surface in CT image. (**a**) Histogram of the CT image volume (**b**) A slice image of CT volume before applying the thresholding. The zoomed in CT image of white box was used in (**c**–**e**). (**c**–**e**) The binary image after thresholding is 1021 to 3000, 1155 to 3000, and 1819 to 3000, respectively.

**Figure 3 bioengineering-11-00027-f003:**
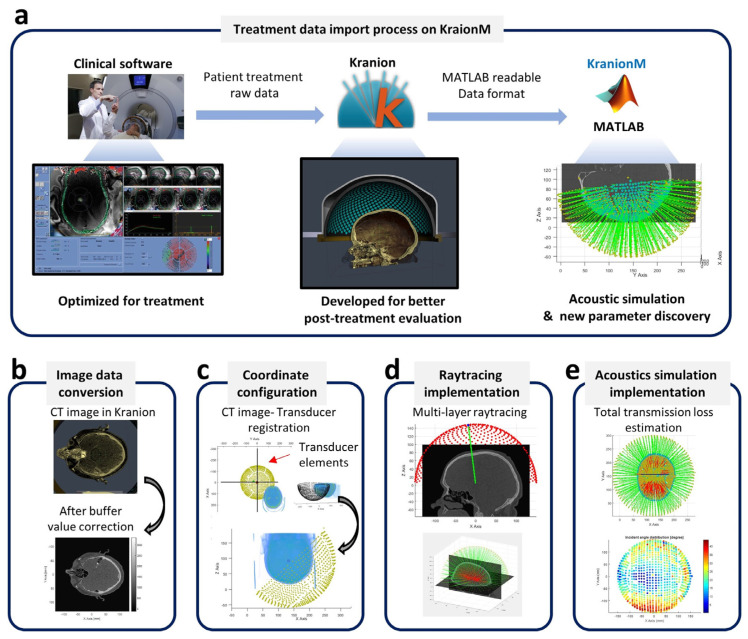
(**a**) The schematic of patient treatment data loading and (**b**–**e**) The key components to implement the raytracing-based acoustic simulation. The image of user interface of clinical system (**a**) is courtesy of Insightec.

**Figure 4 bioengineering-11-00027-f004:**
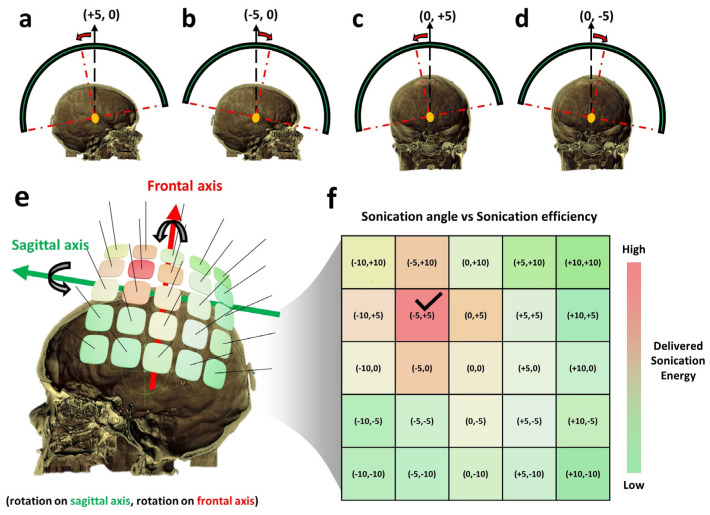
The illustration of FUS transducer titling is on the frontal axis (**a**,**b**) and on the sagittal axis (**c**,**d**). The illustration of a possible rotation angle 2D matrix was rendered on top of the patient’s skull. (**e**) The simulation result was performed on each rotation angle combination. The rotation angle with the highest delivered sonication energy was marked using the check sign on (**f**), and the color of the rectangular brick in (**e**) was also rendered to represent the relative position of the optimum transducer tilting to the skull shape.

**Figure 5 bioengineering-11-00027-f005:**
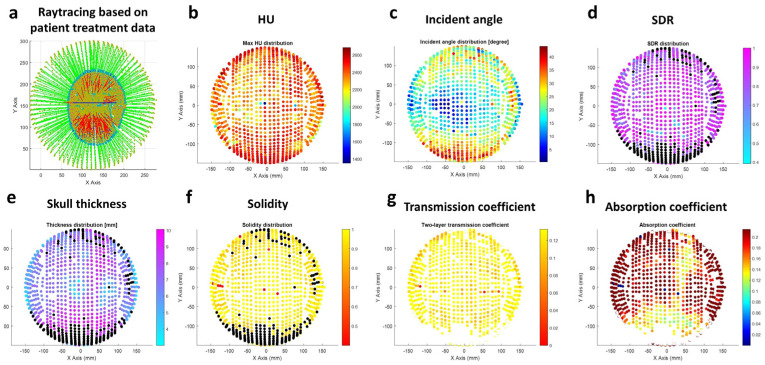
Channel-specific parameters are counted in raytracing. (**a**) A typical snapshot of raytracing simulation based on patient data. The green line indicates the incident beam in the water medium, and the red line indicates the refracted beam in the brain medium through the 3-layered skull. The channel-specific parameters of (**b**) Hounsfield Unit (HU) (**c**) incident angle on outer skull layer (**d**) skull density ratio (SDR) (**e**) skull thickness (**f**) solidity (**g**) combined transmission coefficient on outer and inner skull layer (**h**) absorption coefficient.

**Figure 6 bioengineering-11-00027-f006:**
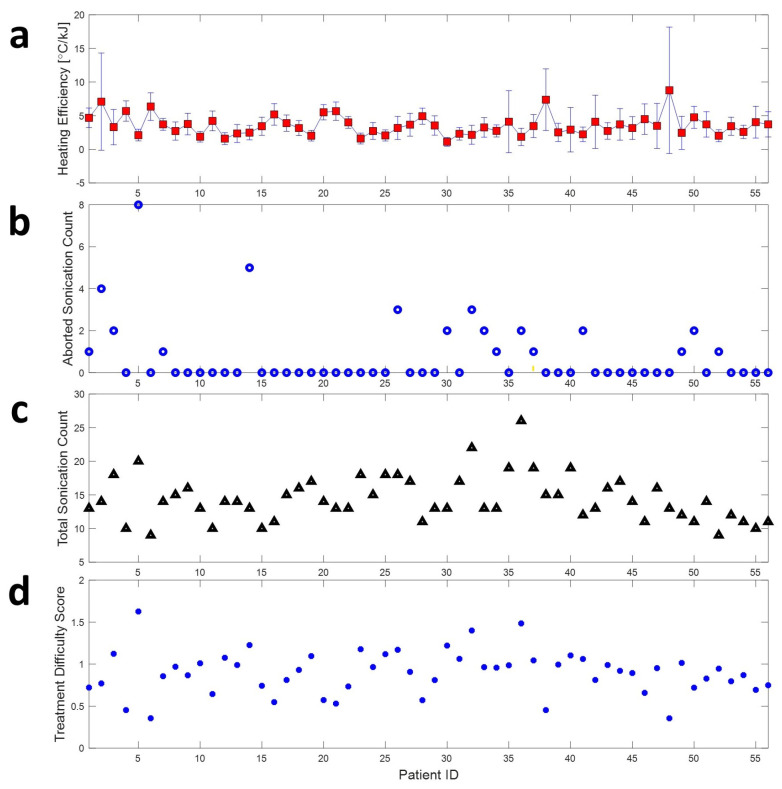
The factors affect the difficulty of the FUS treatment. (**a**) Heating efficiency is calculated by dividing the raised temperature by sonication energy. (**b**) The count of aborted sonication for each treatment session. (**c**) The total count of sonication on each treatment. (**d**) The treatment difficulty score (TDS) takes into account all the abovementioned factors.

**Figure 7 bioengineering-11-00027-f007:**
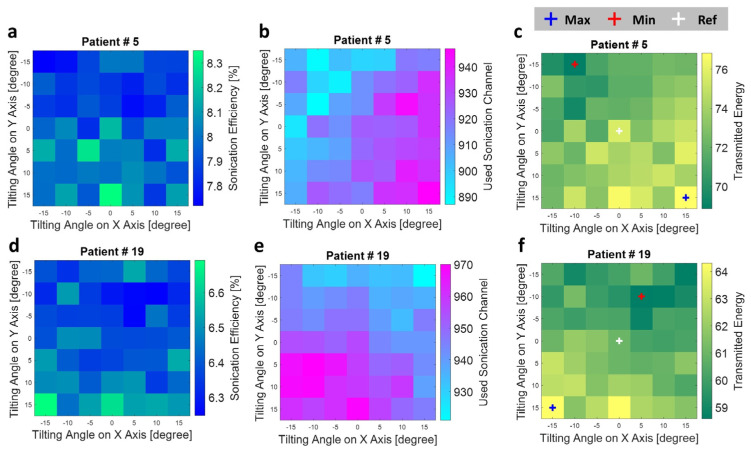
The (**a**) sonication efficiency, (**b**) used sonication channel, and (**c**) transmitted energy based on patient data #5. The (**d**) sonication efficiency, (**e**) used sonication channel, and (**f**) transmitted energy based on patient data #19. The transmitted energy on reference tilting angle (0,0), which is the original treatment setup without transducer tilting, was marked using a white cross, the maximum transmitted energy case was marked with a blue cross, and the minimum transmitted energy case was marked by the red cross in (**c**,**e**).

**Figure 8 bioengineering-11-00027-f008:**
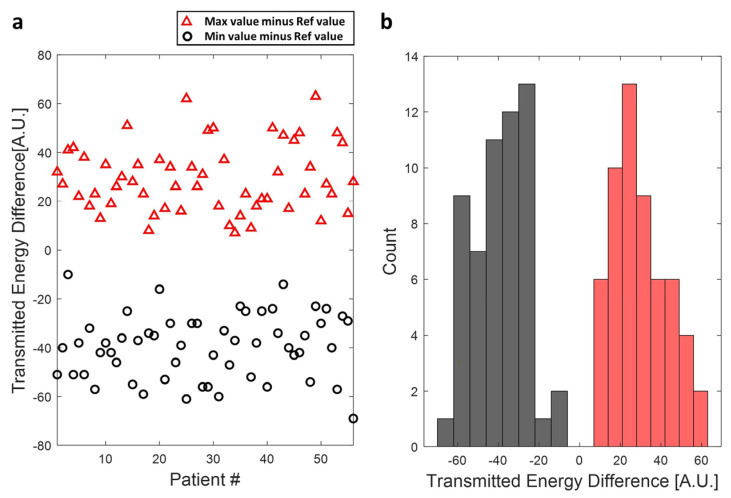
(**a**) The potential improvement in delivered sonication energy of 56 treatment by transducer tilting. The maximum improved delivered energy related to the reference case, which is the original treatment data, was marked using red triangle markers, and the reduced delivered energy compared to the reference case was marked using black circle markers. (**b**) Histogram of potential maximum improvement (red bars) and decrease (black bars) in delivered sonication energy depends on the transducer tilting angle.

**Table 1 bioengineering-11-00027-t001:** The correlation between heating efficiency, aborted sonication count, total sonication count, and the treatment difficulty score.

Correlation	Heating Efficiency	Aborted Sonication Count	Total Sonication Count	Treatment Difficulty Score
**Heating efficiency**	1	-	-	-
**Aborted sonication count**	−0.16	1	-	-
**Total sonication count**	−0.33	0.28	1	-
**Treatment difficulty score**	−0.82	0.55	0.7	1

Note that the value symmetric to the diagonal direction was marked with ‘-’ sign.

## Data Availability

Data are contained within the article.

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
