# Peer review of "Improving Sonication Efficiency in Transcranial MR-Guided Focused Ultrasound Treatment: A Patient-Data Simulation Study"

_bioengineering, 2023, doi:10.3390/bioengineering11010027_

Round 1
Reviewer 1 Report
Comments and Suggestions for Authors
The manuscript presents a comprehensive study on the optimization of transcranial focused ultrasound (TcFUS) treatment through transducer tilting. TcFUS has emerged as a promising modality for treating various brain diseases, including tumors, neuropathic pain, essential tremor, Parkinson's disease, and blood-brain barrier opening. The study focuses on the ExAblate Neuro TcFUS system developed by Insightec and approved by the U.S. Food and Drug Administration (FDA) for treating essential tremor and Parkinson's disease. The complex nature of the human skull, with its heterogeneous and viscoelastic properties, poses a challenge to TcFUS applications. The manuscript proposes a ray tracing-based simulation approach to estimate sonication efficiency for various transducer-patient positions, considering skull porosity, region-specific skull density, and incident angle.
Critical Review: The manuscript provides a detailed and well-structured investigation into the optimization of TcFUS treatment through transducer tilting. The study addresses a crucial aspect of TcFUS by considering the challenges posed by the human skull's complex structure. The use of ray tracing for simulating sonication efficiency, accounting for skull properties and incident angles, adds a valuable dimension to the optimization process. The inclusion of 56 patient treatment cases adds real-world relevance to the study.
Strengths:
- Innovative Approach: The use of ray tracing for sonication efficiency estimation, considering patient-specific skull properties and incident angles, represents an innovative and practical approach.
- Clinical Relevance: The inclusion of patient treatment data enhances the study's clinical relevance, providing insights into potential improvements in TcFUS treatments.
- Comprehensive Analysis: The manuscript thoroughly explores various parameters, such as Hounsfield Unit, incident angle, skull density ratio, and others, providing a comprehensive analysis of the factors influencing sonication efficiency.
Weaknesses:
- Experimental Validation: The proposed optimization method is limited to simulation studies. Experimental validation with physical TcFUS setups would strengthen the study's applicability to real-world scenarios.
- Limited Discussion on Limitations: While the manuscript mentions certain limitations, a more extensive discussion on the challenges and potential shortcomings of the proposed method would enhance the reader's understanding.
- Clarity in Terminology: The manuscript should ensure consistent and clear usage of terminology, especially in defining parameters like sonication efficiency, treatment difficulty score (TDS), and others.
Overall, the manuscript contributes valuable insights into the optimization of TcFUS treatments, offering a promising avenue for future research and potential clinical applications. Addressing the mentioned weaknesses and providing a more nuanced discussion of limitations would further strengthen the manuscript.
Reviewer 2 Report
Comments and Suggestions for Authors
This manuscript investigates potential sonication efficiency improvement achieved by tilting the focused ultrasound transducer in transcranial MR-guided focused ultrasound system, through computer simulations. The major concern is about the simulation methods. The authors use a ray-tracing method, which is far less accurate than full-waveform methods.
1. This is a simulation study, where the Kranion framework by the focused ultrasound foundation is employed. However, this framework is ray-tracing based and far less accurate than full-waveform approaches, e.g., k-Wave. Why not use full-waveform approaches?
2. Please provide key results in the abstract.
3. Please clarify the motivation of this study in the Introduction.
4. How to evaluate the performance of the simulation method?
5. It is better to compare the method with other simulation methods.
6. Please follow the formats of references required by the journal.
Comments on the Quality of English LanguageMinor improvement needed.
Round 2
Reviewer 2 Report
Comments and Suggestions for Authors
Thanks for the revision, which has addressed my concerns.